# Once-Daily Versus Four-Times-Daily Intravenous Busulfan with Therapeutic Drug Monitoring as Conditioning for Hematopoietic Cell Transplantation in Children

**DOI:** 10.3390/pharmaceutics17081081

**Published:** 2025-08-21

**Authors:** Safaa Bazbaz, Irina Zaidman, Ehud Even-Or, Polina Stepensky, Razan Sakran, Daniel Kurnik, Gefen Aldouby-Bier

**Affiliations:** 1Department of Clinical Pharmacy, School of Pharmacy, Faculty of Medicine, Hebrew University of Jerusalem, P.O.Box 12272, Jerusalem 9112102, Israel; 2Hadassah Medical Center, Faculty of Medicine, Department of Bone Marrow Transplantation and Cancer Immunotherapy, Hebrew University of Jerusalem, P.O.Box 12000, Jerusalem 91120, Israel; evenor@hadassah.org.il (E.E.-O.);; 3Clinical Pharmacology and Toxicology Section, Rambam Health Care Campus, P.O.Box 9602, Haifa 3109601, Israel; 4Rappaport Faculty of Medicine, Technion–Israel Institute of Technology, Haifa 3200003, Israel

**Keywords:** busulfan, conditioning therapy, hematopoietic stem cell transplantation, pediatrics, sinusoidal obstruction syndrome, therapeutic drug monitoring, model-informed precision dosing, pharmacokinetics

## Abstract

**Background/Objectives:** Busulfan is a key component of myeloablative conditioning regimens in hematopoietic stem cell transplantation (HSCT) for pediatric patients with acute myeloid leukemia, solid tumors, and certain non-malignant diseases. This study compares the clinical outcomes of once-daily (BU1) versus four-times-daily (BU4) busulfan dosing regimens in pediatric HSCT recipients. **Methods:** A retrospective analysis was conducted on 70 pediatric patients who underwent HSCT at Hadassah Medical Center between June 2018 and October 2023. Thirty-five patients received the BU4 regimen, and 35 received BU1. The primary endpoint was 100-day event-free survival (EFS). **Results:** There was no statistically significant difference in 100-day event-free survival between the BU1 group (88.6%) and the BU4 group (85.7%; *p* = 0.768). Similarly, no significant differences were found in time to neutrophil engraftment (*p* = 0.251) or platelet engraftment (*p* = 0.688). Sinusoidal obstruction syndrome (SOS) occurred in 17.1% of patients in each group. No significant differences were observed in the increase in liver enzyme levels (*p* = 1.0). The incidence of acute graft-versus-host disease was comparable between the groups (41.9% for BU1 vs. 40.0% for BU4; *p* = 0.878). **Conclusions:** Once-daily and four-times-daily busulfan regimens demonstrated comparable clinical outcomes in terms of efficacy and adverse events. Further prospective studies are needed to validate these findings.

## 1. Introduction

Busulfan is a chemotherapeutic agent classified as a bifunctional alkylating compound. It works by alkylating deoxyribonucleic acid (DNA), causing strand breaks and forming crosslinks between DNA strands. These actions interfere with DNA replication and ribonucleic acid (RNA) transcription. Busulfan is not specific to any phase of the cell cycle [1,2,3,4].

It is commonly used in conditioning regimens before hematopoietic stem cell transplantation (HSCT), often in combination with other chemotherapeutic agents and/or radiotherapy [1,2,5,6]. Pediatric indications include acute myeloid leukemia, solid tumors, and certain non-malignant diseases [7]. High-dose busulfan regimens are frequently chosen because of busulfan’s potent cytotoxic effects on hematopoietic stem cells; however, it has a narrow therapeutic index and significant between-subject pharmacokinetic variability, necessitating careful individualized dosing [7].

A multicenter, retrospective cohort analysis by Bartelink et al. suggested that achieving an area under the concentration–time curve (AUC) of 1188–1538 μmol·min (4.9–6.3 mg·h/L) during a six-hour dosing cycle is associated with better event-free survival and fewer acute toxicity events compared with AUCs outside this optimal range [8]. This and other studies indicate that the target exposure for busulfan in myeloablative conditioning regimens should be 4.9 or 5.6 mg·h/L for every six-hour dosing interval, corresponding to cumulative exposures of 19.75 or 22.5 mg·h/L over 24 h (78.8 or 90 mg·h/L over a full four-day regimen), depending on the indication [6,9]. These established targets, combined with busulfan’s narrow therapeutic index and high pharmacokinetic variability, have prompted the adoption of personalized dosing. Once busulfan therapy is initiated, concentration-based dose individualization—implemented as model-informed precision dosing (MIPD)—is recommended to adjust subsequent doses and achieve the desired cumulative exposure [7,10,11].

Busulfan was initially introduced in oral tablet form, but this route had drawbacks, including unpredictable absorption and a high emetogenic potential that often led to drug loss through vomiting [4,5,11]. To improve consistency, intravenous (IV) busulfan was approved by the U.S. Food and Drug Administration in 1999, providing more stable drug concentrations, though individual pharmacokinetic variability remains high [7,12,13]. The standard IV regimen has traditionally involved dosing every six hours (Q6H) as a two-hour infusion over four days, as approved by both the EMA and FDA [6,7].

Reports of accidental oral busulfan overdoses resulting in seizures without other serious events suggested that a more convenient once-daily dose might be well tolerated in children [14,15]. In support of this possibility, Russell et al. conducted a single-arm study involving 70 patients aged 15–64 years to evaluate the clinical toxicity and pharmacokinetics of once-daily IV busulfan; they found that the regimen was convenient, well tolerated, and yielded predictable plasma concentrations [15]. A non-controlled, single-center study by Zwaveling et al. focused on the pharmacokinetics and clinical outcomes of once-daily IV busulfan in 18 children aged 0.5–16 years. After initial dosing based on body surface area, doses were adjusted to achieve a target AUC_0–24_ of 15.6 mg·h/L. Donor engraftment was achieved in 78% of the cases, and disease-free survival was 66% over a median follow-up of 1.6 years. The treatment was generally well tolerated, with minimal toxicity [16].

Park et al. retrospectively examined nine children with acute myelogenous leukemia who underwent HSCT using once-daily IV busulfan [17]. The regimen demonstrated acceptable toxicity, no central nervous system (CNS) events, a low incidence of sinusoidal obstruction syndrome (SOS), and outcomes comparable to standard regimens, suggesting its potential utility in pediatric HSCT [17].

In another retrospective analysis, Bartelink et al. compared outcomes among 60 children undergoing HSCT [18]. Thirty patients received once-daily intravenous busulfan with therapeutic drug monitoring, targeting an AUC_0–24_ of 17.5 mg·h/L per day, while the other 30 received weight-based oral busulfan without concentration-guided dosing. The once-daily IV group showed significantly better event-free survival (83% vs. 30%) and overall survival (83% vs. 53%) compared with the oral group, but also a higher incidence of SOS. These findings suggest that once-daily, concentration-guided IV busulfan may be more effective and could be preferred over non-personalized oral busulfan for pediatric HSCT, particularly in patients at high risk of graft failure or relapse [18].

Until September 2020, the pediatric bone marrow transplant department at Hadassah Medical Center used a busulfan regimen of four intravenous doses per day with concentration-based dose adjustments. Based on accumulating evidence, the protocol was changed to a once-daily dosing schedule, also with concentration-guided adjustments. The current study aims to compare the clinical efficacy and safety of these two IV busulfan dosing regimens in pediatric patients undergoing HSCT. Specifically, it evaluates whether once-daily dosing is non-inferior to the four-times-daily regimen in terms of safety and effectiveness.

## 2. Materials and Methods

This study included 70 patients who underwent HSCT at the Department of Bone Marrow Transplantation and Cancer Immunotherapy, Hadassah Medical Center, Jerusalem, between June 2018 and October 2023. It included all patients who received once-daily busulfan (BU1) after the protocol change in September 2020, as well as the last 35 consecutive patients who had received the traditional four-times-daily protocol (BU4) before the change. Children who did not complete the four-day busulfan conditioning regimen were excluded. HSCT indications were categorized into malignant and non-malignant disorders (see Appendix A). This study was approved by the Hadassah Medical Center Institutional Review Board (HMO-0093-23).

Data on patient and transplant characteristics were collected from hospital medical records, and transplant outcomes were recorded from the first dose of busulfan through the last follow-up. For the BU4 protocol, busulfan was initiated at weight-based doses of 1.1 mg/kg for body weight ≤ 12 kg and 0.8 mg/kg for body weight > 12 kg, consistent with the FDA-approved label. For the BU1 protocol, initial dosing followed a model-based nomogram developed by Bartelink et al. [19].

Blood samples were taken to determine busulfan plasma concentrations using an extensive sampling strategy: nine samples were collected for BU1 patients during and after the first dose, and ten samples for BU4 patients during and after the second dose. Plasma concentrations were measured with a validated liquid chromatography–tandem mass spectrometry assay using a Waters Arc Premier liquid chromatography system coupled to a Xevo TQ-S micro tandem mass spectrometer (Waters, Milford, MA, USA) at the Rambam Health Care Campus referral Laboratory for Clinical Pharmacology, Toxicology, and Pharmacogenetics. Pharmacokinetic parameters were estimated via model-informed precision dosing (MIPD) using the NextDose platform (https://www.nextdose.org/, version 1.4.2 to 2.4.00), based on a previously published population pharmacokinetic model [20]. The target cumulative AUC over the 4-day treatment period ranged from 70 to 90 mg·h·L^−1^, depending on the indication for transplantation and the year of the procedure. Specifically, during the study period, most patients—including those with malignant diseases or inborn errors of immunity—had a target cumulative AUC of 79 mg·h·L^−1^. Beginning in the fall of 2020, the target was increased to 90 mg·h·L^−1^ for patients with metabolic disorders. Conversely, for a small subset of patients with rare non-malignant hematological diseases, such as chronic granulomatous disease, the target cumulative AUC was set at 70 mg·h·L^−1^.

Dose recommendations derived from the model predictions were discussed by the clinical pharmacology team with the attending hemato-oncologist, who decided on any dose adjustments. In general, repeat sampling was performed on the next treatment day when dose changes greater than 25% were recommended.

The primary outcome was event-free survival (EFS) up to 100 days after HSCT. EFS was defined as the time from the first dose of busulfan to the occurrence of any of the following events: relapse in malignant conditions, disease progression in non-malignant conditions, graft failure in allogeneic HSCT recipients, or death from any cause.

Secondary outcomes included time to primary hematopoietic engraftment within 60 days—neutrophil engraftment was defined as the first of three consecutive days with an absolute neutrophil count > 500 µL after the nadir, and platelet engraftment as platelets ≥ 20,000 µL maintained for seven days after the last transfusion; the incidence of acute graft-versus-host disease (aGVHD); busulfan dose adjustments—categorized as increase, decrease, or none—based on model-informed precision dosing at the first sampling and whether a further adjustment was required at the second sampling; the total percentage change in busulfan dose across both sampling occasions; and the incidence of treatment-related adverse events, including sinusoidal obstruction syndrome (SOS) and elevations in liver enzyme levels up to 100 days after the first busulfan dose, recorded as either “no change” or “increased.”

Liver enzyme levels were assessed at baseline, before the initiation of busulfan conditioning, and within 100 days after the first busulfan dose. The tests included serum alkaline phosphatase (ALK), alanine transaminase (ALT), and aspartate transaminase (AST), and results were classified as normal or increased (above the upper limit of normal). Treatment-related adverse events were recorded during the first 90 days after the first dose, focusing on clinically significant and common events as identified from sources such as Pfizer’s medical information site, prescribing information leaflets, UpToDate (accessed on 29 March 2025), and Micromedex (accessed on 29 March 2025) [6,13,21,22]. Potential drug–drug interactions were identified using prescribing information, interaction databases (e.g., drug interaction checkers), the relevant literature, and case reports, and were monitored for up to four half-lives after the final busulfan dose.

All statistical analyses were carried out using IBM SPSS Statistics (version 28.0). Event-free survival (EFS) within the first 100 days post-transplantation was compared between the once-daily and four-times-daily busulfan regimens using Kaplan–Meier survival curves, with differences evaluated via log-rank tests. To assess the association between dosing regimen and EFS while accounting for potential confounders, we employed a multivariable Cox proportional hazards regression model. Covariates in this model included age group, body surface area (BSA), dosing regimen (BU1 vs. BU4), and diagnosis (malignant vs. non-malignant). We also performed a univariable analysis across five diagnostic categories. Hazard ratios (HRs), along with 95% confidence intervals (CIs) and *p*-values, were reported for all variables.

For secondary outcomes, Kaplan–Meier curves were constructed to compare time to primary stem cell engraftment between the once-daily and four-times-daily dosing groups. Differences in these time-to-event outcomes were evaluated using log-rank tests. Categorical outcomes—including the incidence of sinusoidal obstruction syndrome (SOS), acute graft-versus-host disease (aGVHD), elevated liver enzymes, and the need for busulfan dose adjustments based on model-informed precision dosing (MIPD)—were analyzed using chi-square tests (or Fisher’s exact tests when expected cell counts were low). Continuous variables were assessed using independent samples *t*-tests; specifically, the total percentage dose change after MIPD was compared between dosing groups. Normality of dose change percentages was confirmed, and Levene’s test indicated equal variances across groups, justifying the assumption of equal variances in the *t*-test.

Descriptive statistics were reported as means ± standard deviations (SD) for normally distributed continuous variables, medians with interquartile ranges (IQR) for non-normally distributed variables, and frequencies with percentages for categorical variables. A *p*-value < 0.05 was considered statistically significant for all tests.

## 3. Results

### 3.1. Patients and Transplant Characteristics

From June 2018 through October 2023, 70 pediatric patients underwent HSCT with busulfan-based myeloablative conditioning. Thirty-five received the four-times-daily (BU4) protocol, and 35 received the once-daily (BU1) protocol. Detailed patient and transplant characteristics are presented in Table 1 and Table 2.

Non-malignant diseases were the most common indication for HSCT across both groups. Within this category, primary immunodeficiencies were more frequent in the BU4 group, whereas metabolic disorders predominated in the BU1 group. The busulfan–fludarabine (BU–FLU) regimen was the dominant conditioning protocol in both cohorts. Most patients underwent allogeneic HSCT, with bone marrow serving as the primary stem cell source. Fully matched donors were more common than haploidentical or mismatched donors. Most allogeneic transplant recipients received immunosuppressive therapy, primarily antithymocyte globulin (ATG) or alemtuzumab. Graft-versus-host disease prophylaxis, most commonly a combination of cyclosporine and mycophenolate mofetil (CSA–MMF), was administered to all patients undergoing allogeneic HSCT.

### 3.2. Event-Free Survival

During the 100-day follow-up period, four events occurred in the BU1 group—all graft failures (three primary and one secondary). In the BU4 group, five events were observed: three graft failures (two primary and one secondary) and two deaths (one due to septic shock with respiratory failure and one in a patient with refractory acute myeloid leukemia (AML) complicated by severe sinusoidal obstruction syndrome). The 100-day event-free survival (EFS) rate was 88.6% for BU1 and 85.7% for BU4. A log-rank (Mantel–Cox) test showed no statistically significant difference in EFS between the two dosing regimens (χ^2^ = 0.087, *p* = 0.768). Mean EFS times were comparable: 92 days (95% CI = 85–100) for BU1 and 93 days (95% CI = 86–100) for BU4 (Figure 1).

We performed a multivariable Cox proportional hazards analysis to evaluate the associations of dosing frequency, age group, body surface area (BSA), and diagnosis type (malignant vs. non-malignant) with event-free survival (EFS). Dosing frequency was not significantly associated with EFS (hazard ratio [HR] = 0.944; 95% CI = 0.598–1.491; *p* = 0.805). The age group overall showed no significant relationship with EFS (*p* = 0.284). Relative to the reference group of children aged 2–6 years, children aged 2 months–2 years had a non-significant HR of 0.432 (95% CI = 0.036–5.194; *p* = 0.508), while those aged 6–18 years had a non-significant HR of 3.201 (95% CI = 0.234–43.710; *p* = 0.383). A higher BSA appeared to be associated with a lower hazard of events (HR = 0.021; 95% CI = 0.000–2.722), but this did not reach statistical significance (*p* = 0.119). Diagnosis type (malignant vs. non-malignant) was likewise not significantly associated with EFS (HR = 1.939; 95% CI = 0.197–19.091; *p* = 0.570). In a univariable analysis across the five diagnostic categories, no significant association with EFS was observed (HR = 1.218; 95% CI = 0.327–4.537; *p* = 0.769).

### 3.3. Stem Cell Engraftment

#### 3.3.1. Neutrophil Engraftment

Neutrophil engraftment was achieved in 32 out of 35 children (91.4%) in the BU1 group and 33 out of 35 children (94.3%) in the BU4 group. The median time to neutrophil engraftment was 17 days (interquartile range [IQR] = 8 days; range = 10–37) for BU1 and 16 days (IQR = 7 days; range = 7–47) for BU4. Kaplan–Meier analysis (Figure 2) illustrated the cumulative probability of achieving neutrophil engraftment over time for both dosing regimens. A log-rank (Mantel–Cox) test indicated no statistically significant difference in time to neutrophil engraftment between the two groups (χ^2^ = 1.319; *p* = 0.251).

#### 3.3.2. Platelet Engraftment

Within 60 days, platelet engraftment was achieved in 27 of 35 participants (77.1%) in the BU1 group and 26 of 35 participants (74.3%) in the BU4 group. The median time to platelet engraftment was 29 days (IQR = 31 days) for BU1 and 28 days (IQR = 31 days) for BU4. A Kaplan–Meier analysis (Figure 3) showed the cumulative probability of achieving platelet engraftment over the 60-day period for each dosing group. The log-rank test indicated no statistically significant difference between the two regimens (χ^2^ = 0.161; *p* = 0.688).

### 3.4. Incidence of Acute Graft-Versus-Host Disease (aGVHD)

Among patients who underwent allogeneic HSCT, 13 of 31 individuals (41.9%) in the once-daily (BU1) group developed aGVHD, compared with 12 of 30 individuals (40.0%) in the four-times-daily (BU4) group. The difference between groups was not statistically significant (χ^2^ = 0.024; *p* = 0.878).

### 3.5. Dose Adjustments of Busulfan Based on MIPD

After the first pharmacokinetic assessment (Figure 4), 31 of 35 patients (88.6%) in the once-daily (BU1) group required a dose change: 15 (42.9%) needed an increase and 16 (45.7%) needed a decrease. In the four-times-daily (BU4) group, 27 of 35 patients (77.1%) required a dose adjustment, with 21 (60.0%) requiring an increase and six (17.1%) requiring a decrease.

Repeat sampling for model-informed precision dosing was performed in eight patients (22.9%) in the BU1 group and 19 patients (54.3%) in the BU4 group. After this second assessment (Figure 5), six BU1 patients needed an additional adjustment (three increases and three decreases), while eight BU4 patients required a change (six increases and two decreases). Because of the small numbers involved, Fisher’s exact test was used, revealing no statistically significant difference between the groups in the proportion of patients needing a dose change after the second assessment (*p* = 0.209, two-sided).

For patients in the BU4 group, the mean percentage increase in dose was 40.4% (range = 8.33–74.42%), and the mean percentage decrease was −20.8% (range = −5.56% to −35%). In the BU1 group, the mean percentage increase was 18.47% (range = 7–59%), and the mean percentage decrease was −19.63% (range = −4.54% to −49.21%). An independent samples *t*-test revealed a statistically significant difference between the two groups (*t* = −3.157; *p* = 0.002), indicating that patients in the BU4 regimen required greater dose adjustments following model-informed precision dosing than those in the BU1 regimen. Figure 6 presents a waterfall plot illustrating the cumulative percentage dose change per patient after two MIPD adjustments.

### 3.6. Treatment-Related Adverse Events

Sinusoidal obstruction syndrome (SOS) occurred in six patients (17.1%) in each of the BU1 and BU4 groups (χ^2^ = 0; *p* = 1.0), indicating no difference between dosing regimens. Elevated liver enzymes were observed in 15 patients (42.9%) in each group (χ^2^ = 0; *p* = 1.0), likewise showing no significant difference. Other busulfan-related side effects are summarized in Table 3. Overall, adverse events were common in both groups, with some variation in their frequency and type.

One child in the BU1 group experienced three seizures, though these were unlikely to be attributable to busulfan. The patient had a history of seizures and was on chronic valproic acid (VPA) therapy. VPA was discontinued before the conditioning protocol and replaced with benzodiazepines. The seizures occurred approximately 24 h after the last busulfan dose, coinciding with the reinitiation of VPA (including a loading dose and transition to maintenance therapy) and discontinuation of clonazepam—only a single PRN dose was given during the transition. There was minimal overlap between the two antiepileptics, and the seizures took place around the time of antithymocyte globulin (ATG) administration. Moreover, the patient was receiving meropenem, an antibiotic known to reduce VPA plasma concentrations through drug–drug interactions. Together, these factors contributed to the seizures, rather than busulfan itself.

### 3.7. Drug-Drug Interactions

Co-medications with potential to interact with busulfan were observed in both dosing groups. Potentially interacting drugs were identified in 10 patients (28.5%) in the BU1 group and 11 patients (31.4%) in the BU4 group. These medications included paracetamol (five cases in each group), itraconazole (three cases in BU4 and two in BU1), levetiracetam (one in BU4 and two in BU1), and deferasirox (one in BU4).

## 4. Discussion

This study compared two intravenous busulfan dosing regimens—once-daily versus the traditional four-times-daily schedule—both employing model-informed precision dosing (MIPD), in pediatric HSCT patients. To our knowledge, this is the first pediatric study to directly evaluate clinical outcomes between these two approaches using MIPD in both groups. We found no difference in event-free survival between the once-daily and four-times-daily regimens and no associations with age, body surface area, or diagnosis in our relatively small but diverse cohort. These findings align with studies by Russell et al. and Zwaveling et al., which showed that once-daily dosing is well tolerated, provides predictable exposure, and does not adversely affect outcomes in pediatric patients [15,16]. Overall, our results suggest that reducing the dosing frequency from four times per day to once per day may improve convenience and comfort for children and caregivers without compromising the efficacy or safety of busulfan therapy.

In our study, the timing of neutrophil and platelet engraftment did not differ significantly between the once-daily and four-times-daily dosing groups, supporting the equivalence of these regimens in facilitating hematopoietic recovery. This finding aligns with prior studies: Bartelink et al. reported engraftment in 100% of patients treated with once-daily intravenous busulfan and therapeutic drug monitoring, compared with 83% among those receiving weight-based oral busulfan [18], and Zwaveling et al. observed successful engraftment in 14 of 18 children in their single-arm study of once-daily IV busulfan [16].

The importance of dose adjustments based on MIPD was another important aspect of this study. MIPD has been crucial for optimizing busulfan exposure, particularly in pediatric populations where pharmacokinetic variability is significant [7,9,10,23,24]. In our study, most patients in both groups required dose adjustments to reach the target AUC, highlighting the necessity of individualizing therapy. Overall, dose adjustments were significantly larger in the BU4 group than in the BU1 group—a difference likely due to the fact that BU4 dosing began with traditional weight-based recommendations, whereas BU1 dosing was derived from a model-based nomogram.

Among patients who underwent repeat sampling after a substantial dose change, additional adjustments were required more frequently in the BU4 group than in the BU1 group, suggesting that the population pharmacokinetic model used for the BU1 regimen provided more precise dose predictions. The longer interval between infusions in the once-daily regimen (21 h versus 4 h in the four-times-daily regimen) may allow for better characterization of post-infusion concentration profiles and more accurate pharmacokinetic assessments. These findings echo prior research, including the study by Zwaveling et al. that underscored the need for personalized dosing in pediatric HSCT patients [16], and Bartelink et al.’s analysis showing that MIPD-guided once-daily IV busulfan dosing yielded better event-free survival than oral busulfan in pediatric patients undergoing HSCT [18].

In terms of safety, both dosing regimens were associated with expected toxicities. We did not observe a difference in the incidence of sinusoidal obstruction syndrome (SOS) between the once-daily and four-times-daily dosing groups. This contrasts with Bartelink et al., who reported a higher incidence of SOS with once-daily intravenous dosing compared with oral dosing [18]; subsequent work by Kim et al. suggested that elevated busulfan peak concentrations (Cmax) might contribute to this increased risk [25]. However, the higher SOS rate in Bartelink’s study might stem from the lack of concentration monitoring and dose adjustment in the oral busulfan group or from differences in conditioning regimens. Indeed, our and other studies suggest that when guided by MIPD, the BU1 regimen, despite the higher peak concentrations achieved, is not associated with an increased risk for SOS [11]. This is even more remarkable considering the fact that the BU1 group included more patients with metabolic disease whose target cumulative AUC was higher (90 mg*h/L, compared to the standard 79 mg*h/L used in most patients during the study period) and who therefore are expected to achieve even higher peak concentrations.

In our cohort, liver enzyme elevations and acute graft-versus-host disease (aGVHD) occurred at similar rates in both dosing groups. These comparable safety outcomes support the conclusion that once-daily dosing does not increase treatment-related toxicity. Our findings align with those of Ryu et al., who compared once-daily and four-times-daily IV busulfan in adolescents and adults and found no significant difference in adverse effect incidence between the regimens [26].

Overall, adverse event rates were comparable between the once-daily and four-times-daily dosing regimens, with no notable differences observed. Infections were frequent in both groups, underscoring the substantial burden of infectious and immune complications in patients receiving myeloablative conditioning regimens that include busulfan and other agents.

Our study offers several notable strengths. To our knowledge, it is the first pediatric study to compare clinical outcomes between two intravenous busulfan dosing regimens, employing model-informed precision dosing (MIPD) in both groups. We found no significant differences in efficacy or adverse effects between once-daily and four-times-daily dosing, highlighting the value of MIPD in achieving consistent target exposure in pediatric populations with significant pharmacokinetic variability. By comparing the first 35 consecutive patients treated with once-daily dosing after the regimen change with the last 35 patients treated with the four-times-daily regimen before the change, we minimized selection bias.

There are, however, limitations to our work. Firstly, our study was conducted at a single center and involved a relatively small and diverse cohort, which may limit the generalizability of the findings. Our study was a retrospective cohort study with a 5-year recruitment period, and we could not account for potential time-related confounders, such as changes in clinical practice over the study period. For instance, in patients undergoing transplantation for metabolic diseases, in 2020, the target cumulative AUC was increased from the traditional target (79 mg*h/L) to 90 mg*h/L, reflecting accumulating evidence for greater effectiveness. Thus, patients with this higher target were represented in the BU1 group only, and a higher rate of exposure-related adverse effects would have been expected in this group. Nonetheless, treatment-related toxicity was comparable between the two regimens, emphasizing the safety of the BU1 regimen.

Also, in view of our comparatively small sample size, we did not pre-specify non-inferiority margins or perform formal power calculations, since the primary aim was an exploratory analysis of the outcomes after switching to the BU1 regimen. Similarly, with a moderate cohort size, we were restricted in the number of covariates included in multivariable models. For instance, we did not include the conditioning protocol as a covariate. However, all conditioning protocols were myeloablative, and there were only small differences in the distribution of conditioning protocols between the 2 dosing regimens, so that it is unlikely that such differences significantly affected our findings. Nonetheless, in view of the small number of events in the various event categories, our multivariable Cox proportional hazards regression models for various dichotomous outcomes were likely overfitted and should therefore be considered only exploratory. In order to robustly detect differences of modest effect size between the treatment regimens, as well as to comprehensively evaluate the contributions of potential covariates and minimize the influence of residual confounding, conducting a large, multicenter randomized controlled trial would be necessary. However, given the widespread adoption of the once-daily busulfan regimen (BU1), conducting such a trial would likely prove challenging. Lastly, because our focus was on pediatric patients, these results cannot be directly applied to adult populations.

This study provides evidence that once-daily intravenous busulfan dosing is an effective and safe alternative to the traditional four-times-daily regimen for pediatric patients undergoing HSCT. By delivering similar clinical outcomes while offering improved convenience for both patients and clinical staff, the once-daily approach—when coupled with model-informed precision dosing to optimize drug exposure—has strong potential to become the preferred standard in clinical practice.

## Figures and Tables

**Figure 1 pharmaceutics-17-01081-f001:**
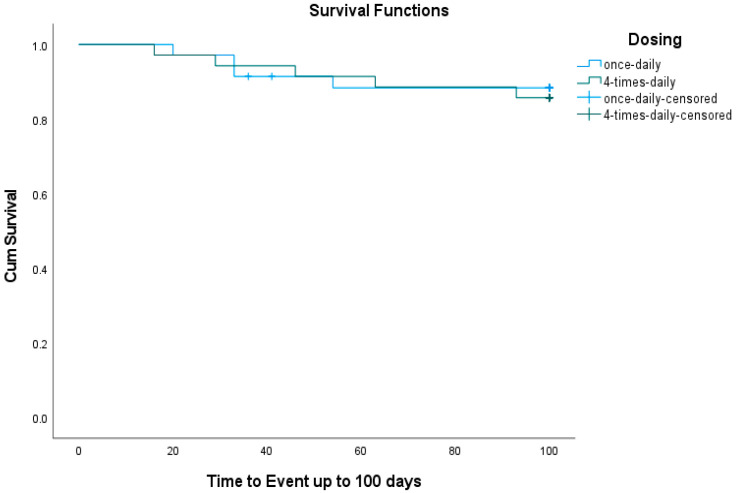
Cumulative event-free survival up to 100 days.

**Figure 2 pharmaceutics-17-01081-f002:**
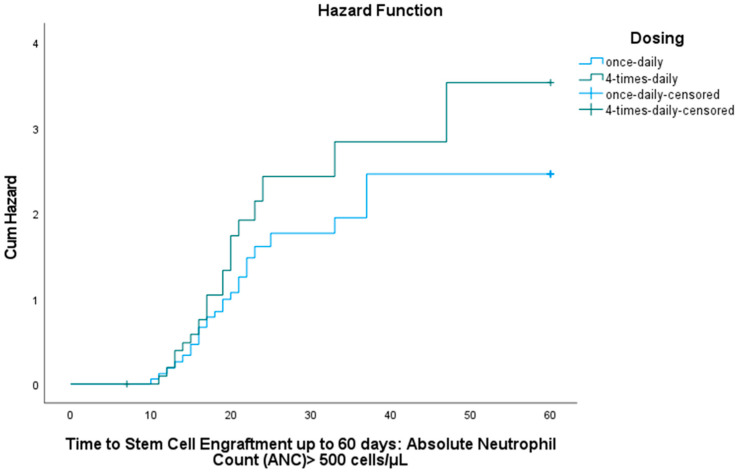
Cumulative hazard of neutrophil engraftment over 60-day period (ANC > 500 cells/mL).

**Figure 3 pharmaceutics-17-01081-f003:**
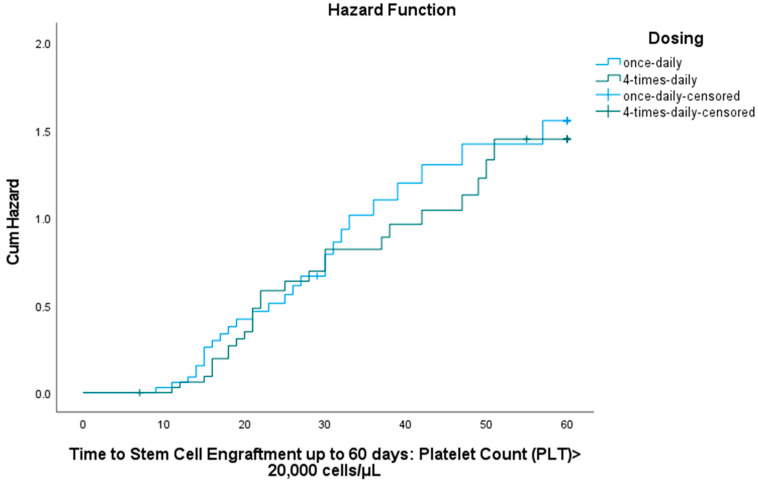
Cumulative hazard of platelet engraftment over 60-day period (PLT > 20,000 cells/mL).

**Figure 4 pharmaceutics-17-01081-f004:**
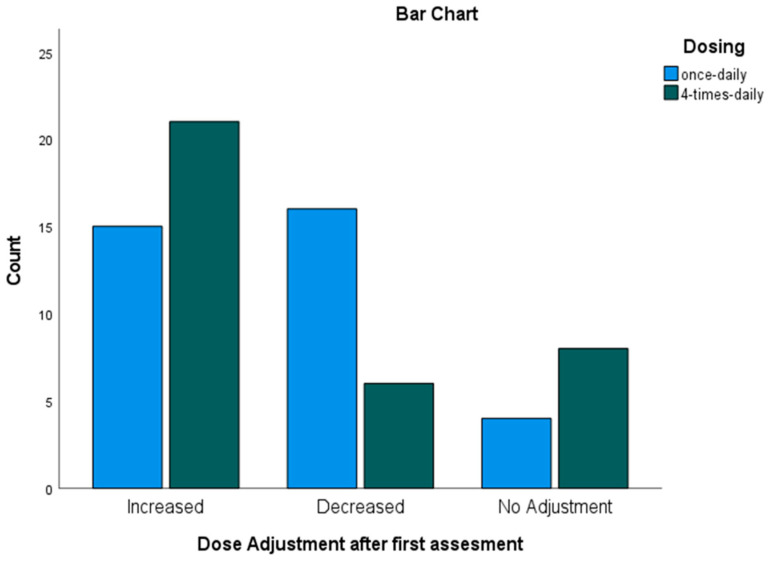
Dose adjustments based on MIPD after the first assessment.

**Figure 5 pharmaceutics-17-01081-f005:**
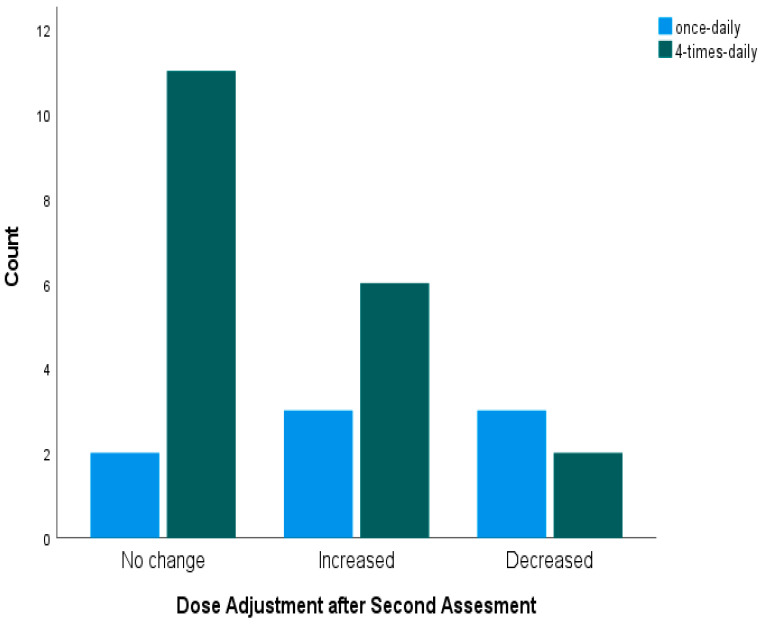
Dose adjustments based on MIPD after the second assessment.

**Figure 6 pharmaceutics-17-01081-f006:**
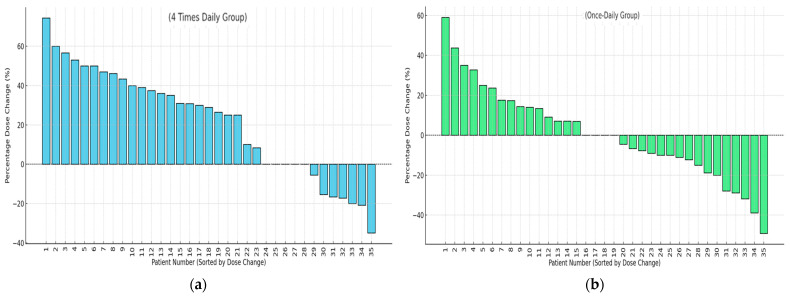
Cumulative percent dose changes per patient after MIPD for the BU4 group (panel (**a**)) and for the BU1 group (panel (**b**)).

**Table 1 pharmaceutics-17-01081-t001:** Patient characteristics.

Patient Characteristics	BU4 *n* = 35 (%)	BU1 *n* = 35 (%)	*p*-Value
**Age at HSCT, median (years)**—(25th–75th Percentile (IQR))	3.85 (1.18–8.29 (7.11))	4.83 (3.08–6.6 (3.52))	0.304
**Follow up, median (days)**—(25th–75th Percentile (IQR))	1169 (343–1587 (1244))	267 (174–537 (363))	<0.001
**Age group (years)**			0.397
<2 years	12 (34.3%)	7 (20%)	
2–<6 years	10 (28.6%)	13 (37.1%)	
6–18 years	13 (37.1%)	15 (42.9%)	
**Sex**			0.131
Male	20 (57.1%)	26 (74.3%)	
Female	15 (42.9%)	9 (25.7%)	
**Body surface area, median (m^2^)**—(25th–75th Percentile (IQR))	0.59 (0.44–0.92 (0.48))	0.72 (0.59–0.93 (0.34))	0.147
**Diagnosis**			0.794 *
Malignant Diseases:	11 (31.4%)	10 (28.6%)	
Leukemia	5	6	
Solid tumors	6	4	
Non-malignant Diseases:	24 (68.6%)	25 (71.4%)	
Inborn errors of immunity	14	6	
Metabolic disorders	5	17	
Non-malignant hematological disorders	5	2	

* Binary variable.

**Table 2 pharmaceutics-17-01081-t002:** Transplant characteristics.

Transplant Characteristics	BU4 *n* = 35 (%)	BU1 *n* = 35 (%)
**Type of HSCT**		
Allogeneic	30 (85.7%)	31 (88.6%)
Autologous	5 (14.3%)	4 (11.4%)
**Conditioning Protocol**		
BU-FLU	20 (57.1%)	20 (57.1%)
BU-MEL	5 (14.3%)	4 (11.4%)
BU-CYC-MEL	5 (14.3%)	5 (14.3%)
BU-FLU-CYC	0 (0%)	5 (14.3%)
BU-FLU-TT	5 (14.3%)	1 (2.9%)
**Allogeneic Patients**	**AlloBU4** ***n* = 30 (%)**	**AlloBU1** ***n* = 31 (%)**
**Donor Source Cells**		
BM	23 (76.7%)	16 (51.6%)
PBSC	6 (20%)	5 (16.1%)
Cord	1 (3.3%)	10 (32.3%)
**Type of Donor**		
Related donor	16 (53.3%)	10 (32.3%)
Unrelated donor	14 (46.7%)	21 (67.7%)
**HLA Matching**		
Full matched donor	23 (76.7%)	23 (74.2%)
Haploidentical	4 (13.3%)	2 (6.5%)
Mismatched donor	3 (10%)	6 (19.4%)
**Immunosuppressive** **Therapy**		
Yes:	27 (90%)	29 (93.5%)
ATG	20	25
Alemtuzumab	7	4
**GVHD Prophylaxis**		
CSA-MMF	17 (56.7%)	22 (70.9%)
CSA-MTX	6 (20%)	6 (19.4%)
CSA only	5 (16.7%)	3 (9.7%)
CSA-MTX-MMF	2 (6.7%)	0 (0%)

Abbreviations: Allo: Allogeneic, ATG: anti-thymocyte globulin, BM: bone marrow, BU: busulfan, CYC: cyclophosphamide, CSA: cyclosporine, FLU: fludarabine, HLA: human leukocyte antigen, MEL: melphalan, MMF: mycophenolate, MTX: methotrexate, PBSC: peripheral blood stem cells, TT: thiotepa.

**Table 3 pharmaceutics-17-01081-t003:** Treatment-related adverse events.

Busulfan Adverse Events	BU4–*n* = 35 (%)	BU1–*n* = 35 (%)
Fever	33 (94.3%)	32 (91.4%)
Headache	6 (17.1%)	3 (8.6%)
Chills	8 (22.9%)	5 (14.3%)
Pain	25 (71.4%)	27 (77.1%)
Edema	6 (17.1%)	4 (11.4%)
Refractory Thrombocytopenia	9 (25.7%)	7 (20.0%)
Tachycardia	5 (14.3%)	3 (8.6%)
Hypertension	7 (20.0%)	5 (14.3%)
Thrombosis	2 (5.7%)	2 (5.7%)
Mucositis	35 (100%)	35 (100%)
Stomatitis	12 (34.3%)	14 (40.0%)
Nausea	15 (42.9%)	17 (48.6%)
Vomiting	14 (40.0%)	16 (45.7%)
Diarrhea	18 (51.4%)	20 (57.1%)
Constipation	11 (31.4%)	13 (37.1%)
Abdominal Pain	6 (17.1%)	8 (22.9%)
Abdominal Enlargement	2 (5.7%)	2 (5.7%)
**Metabolic and Nutritional System**		
Hypomagnesemia	21 (60.0%)	20 (57.1%)
Hypokalemia	18 (51.4%)	17 (48.6%)
Hypocalcemia	4 (11.4%)	5 (14.3%)
Hyperbilirubinemia	3 (8.6%)	3 (8.6%)
Creatinine Increased	7 (20.0%)	6 (17.1%)
**Respiratory System**		
Epistaxis	4 (11.4%)	4 (11.4%)
Dyspnea	6 (17.1%)	7 (20.0%)
**Others**		
Rash	9 (25.7%)	10 (28.6%)
Seizures	0 (0.0%)	1 (2.9%)
**Infections**		
Bacterial Infections	10 (28.6%)	8 (22.9%)
Fungal Infections	6 (17.1%)	4 (11.4%)
Viral Infections	12 (34.3%)	9 (25.7%)

## Data Availability

The data that support the findings of this study are available from the corresponding author upon reasonable request.

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
