# Peer review of "Once-Daily Versus Four-Times-Daily Intravenous Busulfan with Therapeutic Drug Monitoring as Conditioning for Hematopoietic Cell Transplantation in Children"

_pharmaceutics, 2025, doi:10.3390/pharmaceutics17081081_

Round 1
Reviewer 1 Report
Comments and Suggestions for Authors
This manuscript by Bazbaz et al. is well-designed and a valuable addition to the therapeutic monitoring of busulfan. This study is within the scope of Pharmaceuticals. However, several aspects need clarification. The Authors are kindly asked to consider the following comments:
- Target exposure levels for busulfan in children vary depending on the protocols and dosing regimens. The authors mentioned repeated sampling was performed on the following treatment day when dose changes were made. What total AUC value was used as the target exposure in the BU1 and BU4 groups, based on which dose adjustments were made? It is worth adding this data in the materials and methods section.
- The authors should present the AUC values obtained in the study groups in the Results section. This parameter is crucial in the TDM of busulfan.
- In the Discussion section, the authors should comment on the fludarabine used in most patients, which could act as the main confounding factor on the measured clinical endpoint. Were there differences in the clearance values on the following days of busulfan administration? Did the pharmacokinetic model applied take into account this aspect?
- Please explain the reason for collecting as many as 9 samples for BU1 and 10 for BU4 in the pediatric profile. Available models allow for estimation of PK parameters based on just 5, 6, or even 3 blood samples. A limited sampling strategy is crucial, especially in the pediatric population. Was such frequent collection problematic in the case of the BU4 group? Please comment.
- In the Results section, the authors showed that adverse events were comparable in both groups. It would be worth conducting such an analysis in groups depending on the AUC values achieved and whether the incidence of SOS was higher in the subset of patients with a high deviation from the target AUC compared to patients with the correct exposure.
Author Response
Review Report 1 Response
Thank you very much for taking the time to review this manuscript. Please find detailed responses below and the corresponding corrections highlighted in the re-submitted files.
- Target exposure levels for busulfan in children vary depending on the protocols and dosing regimens. The authors mentioned repeated sampling was performed on the following treatment day when dose changes were made. What total AUC value was used as the target exposure in the BU1 and BU4 groups, based on which dose adjustments were made? It is worth adding this data in the materials and methods section.
Response: The target exposure is indeed an important piece of information which we omitted to include. The target cumulative AUC for the 4-day treatment period ranged from 70 to 90 mg·h·L⁻¹, depending on the transplantation indication and the year of transplant. In particular, during the study period, for most patients (e.g., with malignant disease and inborn errors of immunity), the target cumulative AUC was 79 mg·h·L⁻¹, while since the fall of 2020, for patients with metabolic diseases, the target was increased to 90 mg·h·L⁻¹, while for a few patients with rare hematological non-malignant diseases (e.g., chronic granulomatous disease), the target cumulative AUC was 70 mg·h·L⁻¹. In Israel, only during the last 1-2 years have higher target cumulative exposures (e.g., 90 mg·h·L⁻¹) been adopted for some conditioning protocols for patients with malignant diseases.
We have now added the following information to the Methods section:
Methods and Materials, p. 3, line 130: “… based on a previously published population pharmacokinetic model [(McCune et al., 2014)]. The target cumulative AUC over the 4-day treatment period ranged from 70 to 90 mg·h·L⁻¹, depending on the indication for transplantation and the year of the procedure. Specifically, during the study period, most patients—including those with malignant diseases or inborn errors of immunity—had a target cumulative AUC of 79 mg·h·L⁻¹. Beginning in the fall of 2020, the target was increased to 90 mg·h·L⁻¹ for patients with metabolic disorders. Conversely, for a small subset of patients with rare non-malignant hematological diseases, such as chronic granulomatous disease, the target cumulative AUC was set at 70 mg·h·L⁻¹.
- The authors should present the AUC values obtained in the study groups in the Results section. This parameter is crucial in the TDM of busulfan.
The estimated AUC depends on both dose and clearance and will therefore be strongly influenced by the way the first dose is determined. However, the determination of the first dose differed between the BU4 and BU1 groups: While for the BU4 regimen, we used body weight- based dosing recommendation according to the Busulfex label, in the BU1 group, a model-based nomogram developed by Bartelink et al. was used. Therefore, AUCs values after the first monitoring occasion would not be comparable between the BU1 and BU4 dosing regimens, and in our view would be misleading.
However, as detailed in our response to item 1, the dose recommendation provided by the population PK model accounts for both dose and exposure on the first treatment day and therefore integrates all parameters to calculate the dose required to achieve the target cumulative AUC of 79 mg·h·L⁻¹. Dose recommendations provided by the model implemented in Nextdose have excellent bias and precision in achieving the cumulative target AUC (Lawson et al., 2021).
- In the Discussion section, the authors should comment on the fludarabine used in most patients, which could act as the main confounding factor on the measured clinical endpoint. Were there differences in the clearance values on the following days of busulfan administration? Did the pharmacokinetic model applied take into account this aspect?
Response: The two treatment groups were well- balanced regarding fludarabine comedication as part of the conditioning regimen (71.4% of patients in the BU4 group and 74.3% in the BU1 group), and it is therefore unlikely that fludarabine use confounded our findings.
Moreover, although some initial studies reported an interaction between oral busulfan and fludarabine, the consensus nowadays is that there is no clinically significant pharmacokinetic interaction between IV busulfan and fludarabine(Domingos et al., 2024). For instance, among 44 population PK models evaluated, in only 2 models fludarabine comedication retained as a significant covariate on BU clearance after accounting for all other covariates, and the mean effect size was negligible (4%)(Takahashi et al., 2023). Therefore, our MIPD model does not account for fludarabine comedication.
Irrespective of fludarabine co-therapy, most contemporary busulfan models, including the McCune model used in NextDose, incorporate time-varying clearance, with a decline in clearance of 8-14% on the second treatment day.
In summary, the two treatment groups were balanced regarding fludarabine co-therapy, and there is no evidence that fludarabine has a significant effect on busulfan pharmacokinetics. Therefore, fludarabine co-therapy is unlikely to confound our findings.
- Please explain the reason for collecting as many as 9 samples for BU1 and 10 for BU4 in the pediatric profile. Available models allow for estimation of PK parameters based on just 5, 6, or even 3 blood samples. A limited sampling strategy is crucial, especially in the pediatric population. Was such frequent collection problematic in the case of the BU4 group? Please comment.
Response: Until the introduction of BU MIPD in our department around 2017, we had used non-compartmental PK analysis for BU dose adjustments, determining the AUC by trapezoidal rule which requires dense sampling. Therefore, the nursing staff is well accustomed to the sampling, and we usually do not have any technical issues. Since the introduction of MIPD, we have continued using dense sampling, allowing us to refine our BU model in particular with respect to the effect of lag time and flushing time on model performance. However, we will soon adopt a limited sampling approach of 5 samples, using an optimized version of a population PK model. This will certainly reduce the load on patients and staff alike.
- In the Results section, the authors showed that adverse events were comparable in both groups. It would be worth conducting such an analysis in groups depending on the AUC values achieved and whether the incidence of SOS was higher in the subset of patients with a high deviation from the target AUC compared to patients with the correct exposure.
Response: As discussed in our response to Comment 2, MIPD aims to reach the target exposure irrespective of the AUC achieved after the first dose Therefore, all patients are expected to have reached a similar cumulative AUC (around 79 mg*h/L). Indeed, using the McCune model implemented in NextDose, the target AUC can be reached with high precision and low bias (Lawson et al., 2021). Therefore, we do not believe that an analysis of adverse drug events based on the AUC achieved on day 1 would be meaningful. Also, in view of the high between-patient variability in BU clearance, larger cohorts would be necessary to show an association of ADEs with BU exposure.
There is some data linking high peak concentrations of BU with the incidence of SOS. Since the peak concentrations reached with the BU1 regimen are much higher than those reached with the BU4 regime, there was concern that the BU1-regimen would be associated with a higher incidence of SOS. However, these concerns have not been corroborated by our findings and other studies. We have now expanded this point in the Discussion:
Discusson: “We did not observe a difference in the incidence of sinusoidal obstruction syndrome (SOS) between the once‑daily and four‑times‑daily dosing groups. This contrasts with Bartelink et al., who reported a higher incidence of SOS with once‑daily intravenous dosing compared with oral dosing [(Bartelink et al., 2008)]; subsequent work by Kim et al. suggested that elevated busulfan peak concentrations (Cmax) might contribute to this increased risk [(Kim et al., 2024)]. However, the higher SOS rate in Bartelink’s study might stem from the lack of concentration monitoring and dose adjustment in the oral busulfan group or from differences in conditioning regimens. Indeed, our and other studies suggest that when guided by MIPD, the BU1 regimen, despite the higher peak concentrations achieved, is not associated with an increased risk for SOS(Domingos et al., 2024). This is even more remarkable considering the fact that the BU1 group included more patients with metabolic disease whose target cumulative AUC was higher (90 mg*h/L, compared to the standard 79 mg*h/L used in most patients during the study period) and who therefore are expected to achieve even higher peak concentrations.”
References:
Bartelink, I. H., Bredius, R. G. M., Ververs, T. T., Raphael, M. F., van Kesteren, C., Bierings, M., Rademaker, C. M. A., den Hartigh, J., Uiterwaal, C. S. P. M., Zwaveling, J., & Boelens, J. J. (2008). Once-Daily Intravenous Busulfan with Therapeutic Drug Monitoring Compared to Conventional Oral Busulfan Improves Survival and Engraftment in Children Undergoing Allogeneic Stem Cell Transplantation. Biology of Blood and Marrow Transplantation, 14(1), 88–98. https://doi.org/10.1016/j.bbmt.2007.09.015
Domingos, V., Nezvalova-Henriksen, K., Dadkhah, A., Moreno-Martinez, M. E., Ben Hassine, K., Pires, V., Kröger, N., Bauters, T., Hassan, M., Duncan, N., Kalwak, K., Ansari, M., Langebrake, C., & Admiraal, R. (2024). A practical guide to therapeutic drug monitoring in busulfan: recommendations from the Pharmacist Committee of the European Society for Blood and Marrow Transplantation (EBMT). Bone Marrow Transplantation, 59(12), 1641–1653. https://doi.org/10.1038/S41409-024-02413-0,
Kim, Y., Moon, S., & Rhee, S. J. (2024). Optimal Once-Daily Busulfan Administration in Pediatric Patients: A Simulation-Based Investigation of Intravenous Infusion Times. Drug Design, Development and Therapy, 18, 871–879. https://doi.org/10.2147/DDDT.S451970
Lawson, R., Paterson, L., Fraser, C. J., & Hennig, S. (2021). Evaluation of two software using Bayesian methods for monitoring exposure and dosing once-daily intravenous busulfan in paediatric patients receiving haematopoietic stem cell transplantation. Cancer Chemotherapy and Pharmacology, 88(3), 379–391. https://doi.org/10.1007/S00280-021-04288-0,
McCune, J. S., Bemer, M. J., Barrett, J. S., Baker, K. S., Gamis, A. S., & Holford, N. H. G. (2014). Busulfan in infant to adult hematopoietic cell transplant recipients: A population pharmacokinetic model for initial and bayesian dose personalization. Clinical Cancer Research, 20(3), 754–763. https://doi.org/10.1158/1078-0432.CCR-13-1960,
Takahashi, T., Jaber, M. M., Brown, S. J., & Al-Kofahi, M. (2023). Population Pharmacokinetic Model of Intravenous Busulfan in Hematopoietic Cell Transplantation: Systematic Review and Comparative Simulations. Clinical Pharmacokinetics, 62(7), 955–968. https://doi.org/10.1007/S40262-023-01275-X,

Reviewer 2 Report
Comments and Suggestions for Authors
The manuscript explores the feasibility of once-daily versus four-times-daily busulfan dosing using Model-Informed Precision Dosing (MIPD) in pediatric hematopoietic stem cell transplantation. While the study addresses a clinically relevant question, several methodological and interpretive issues need clarification, particularly concerning confounding variables, dosing strategies, statistical power, and follow-up duration. Here are some questions can modify the clarity and rigor of the manuscript:
1. How did the authors adjust for potential confounding due to changes in clinical practice over the five-year enrollment period?
2. Was a non-inferiority margin or power calculation pre-specified, given the small number of EFS events?
3. How was heterogeneity in conditioning backbones controlled when comparing BU1 vs BU4 outcomes?
4. Could differences in initial dosing methods (Bartelink nomogram vs weight-based) account for outcome differences rather than dosing frequency?
5. Given the low event count, were model diagnostics or penalized regression methods used to prevent overfitting?
6. Why was analysis limited to 100-day outcomes despite unequal and longer follow-up in the BU4 group?
7. Were Cmax values analyzed to assess exposure–toxicity relationships, particularly regarding SOS incidence?
8. How were potential drug–drug interactions managed or adjusted for in the analysis of busulfan pharmacokinetics?
Round 2
Reviewer 2 Report
Comments and Suggestions for Authors
Accepted in the present form. I appreciate the authors they revised their manuscript as per reviewer suggestion. However, please thoroughly check the manuscript before publication to prevent any types of errors.